# Economic Aspects of Bovine Ephemeral Fever (BEF) Outbreaks in Dairy Cattle Herds

**DOI:** 10.3390/vetsci10110645

**Published:** 2023-11-08

**Authors:** Yaniv Lavon, Ephraim Ezra, Orly Friedgut, Adi Behar

**Affiliations:** 1Israel Cattle Breeders Association, Caesarea 38900, Israel; yaniv@icba.co.il (Y.L.); ephraim@icba.co.il (E.E.); 2Kimron Veterinary Institute, Bet Dagan 50250, Israel; orlyf@moag.gov.il

**Keywords:** bovine ephemeral fever, bovine ephemeral fever virus, arboviruses, dairy cattle, dairy cattle management, climate changes

## Abstract

**Simple Summary:**

Bovine ephemeral fever (BEF) is a viral disease of cattle that is transmitted by blood-feeding insects. In Israel, farmers routinely report data on every BEF case to the Farm Herd Management Program (NOA), and they are registered in the Israel Cattle Breeders Association herd book. In this study, we used the statistical capability of national data stored in the Israeli herd book to evaluate the economic effects of BEF outbreaks. Our results show substantial economic losses from the reduction in milk production and culling of valuable cows. Due to climatic change, the risk of bovine ephemeral fever virus (BEFV) emergence and spread in Europe is real. Since the European cattle population has never been exposed to BEFV, the economic losses to dairy and beef production in this continent during its first BEF outbreak may be considerable. Additionally, it could also cause financial damage due to restrictions on animal trade and transportation, like the current EHDV-8 outbreak in the Mediterranean basin. These results, exhibiting for the first time to our knowledge, the impact of BEF outbreaks at a population level could enable us to conduct an accurate risk assessment in future cases of BEFV emergence.

**Abstract:**

Bovine ephemeral fever virus (BEFV) is an arthropod-borne virus (arbovirus) transmitted by blood-feeding insects (mosquitoes and *Culicoides* biting midges). While the dispersal of arboviral diseases such as bovine ephemeral fever (BEF) into naive areas is often the result of globalization and animal movement, the endemization and local outbreaks of these diseases are mainly influenced by environmental changes. Climate change affects the activity, distribution, dynamics, and life cycles of these vectors (arthropods), the replication of viruses within their vectors, and weakens animal’s immune systems. Although BEF does not currently occur in the Americas and Europe (other than in the western regions of Turkey), the risk of BEFV emergence, spread, and endemization in Europe is real. Over the past two decades, arboviruses such as the bluetongue virus (BTV) and Schmallenberg virus (SBV) have emerged in Europe without warning and caused significant losses to the dairy and meat industries. Since the European cattle population has never been exposed to BEFV, the economic losses to dairy and beef production in this continent due to the reduction in milk production, loss of valuable cows, and abortion, should BEF emerge, would probably be considerable. Moreover, arboviruses can also cause substantial financial damage due to restrictions on animal trade and transportation, like the current EHDV-8 outbreak in the Mediterranean basin. In this study, we used national data stored in the Israeli herd book to examine the economic aspects of BEF outbreaks in affected dairy cattle farms countrywide. Our results demonstrate that BEF outbreaks can have immediate and delayed effects, causing severe economic losses due to culling (loss of valuable cows) and a reduction in milk production that affects dairy farm income for months after clinical diagnosis. To our knowledge, this is the first extensive study on the impact of a BEF outbreak at a population level, enabling to conduct accurate risk assessments in future cases of BEFV emergence and re-emergence.

## 1. Introduction

Bovine ephemeral fever virus (BEFV) is a rhabdovirus classified as the type species of the genus *Ephemerovirus* (species *Ephemerovirus febris*, genus *Ephemerovirus*, family *Rhabdoviridae*). Similar to other members of this family, BEFV is a negative-sense single-strand (ss) RNA virus with a bullet-shaped morphology [1]. Its 14.9 Kb genome encodes a non-structural glycoprotein (GNS) and five structural proteins as follows: nucleoprotein (N); polymerase-associate protein (P); matrix protein (M); viral RNA polymerase-associate protein (L); and surface glycoprotein (G). The genome also encodes four open-reading frames (ORFs) [2]. BEFV is an arthropod-borne virus (arbovirus) that is suspected to be transmitted by mosquitoes and *Culicoides* biting midges [3,4,5,6].

The disease caused by BEFV is bovine ephemeral fever (BEF). BEF manifests in anorexia, depression, ocular and nasal discharge, salivation, muscle stiffness, lameness, rumenal stasis, sternal recumbency, and other inflammatory responses [3,7,8,9]. This viral disease is also known as a three-day sickness due to the spontaneous recovery of the cattle within three days. BEF is an important viral disease of cattle (and the water buffalo) in tropical, subtropical, and temperate climatic zones. It is widely distributed in Africa, Asia (including most of China, Taiwan, the Korean Peninsula, and Japan), the Middle East, and Australia, with seasonal outbreaks occurring from late spring to autumn [3,6,9,10]. Despite its short duration, the disease can cause heavy economic losses due to decreased milk production, lowered fertility in bulls, and even fatality in severe cases [8,11].

While the dispersal of arboviral diseases such as BEF into naive areas is often the result of globalization and animal movement [12], the endemization and local outbreaks of these diseases and their severity are mainly influenced by environmental changes [13]. Climate change affects the activity and distribution of these vectors (arthropods) as well as the length and frequency of the vectors’ life cycles [14,15]. Climate change also affects the replication of viruses in their vectors (mainly temperature) and the dynamics (density and composition) of vector populations (temperature and humidity of both the air and the soil) [13,16]. Climate factors also contribute to a decrease in the resistance of farm animals to infection as a result of heat stress and the weakening of animals’ immune systems [13]. Although BEF does not occur in the Americas and Europe (other than in the western regions of Turkey) [17], the risk of BEFV emergence, spread, and ultimately endemization in Europe due to climatic change is real. In the past two decades, arboviruses such as the bluetongue virus (BTV) and Schmallenberg virus (SBV) have emerged in Europe without warning, causing significant losses to the dairy and meat industries. In 2007 alone, a BTV epidemic in France was estimated to have cost USD 1.4 billion [18]. Belgium estimated the mean cost for SBV-infected animals to be EUR 65 and 107 in the case of a fatal outcome or apparent recovery, respectively [19]. Since the European cattle population has never been exposed to BEFV, the economic losses to dairy and beef production in this continent, should BEF emerge, will probably be considerable. In the eastern Mediterranean basin, Israel has tropical and continental climate conditions which enable the spread and endemization of several arboviruses, including BEFV [20]. BEF was first described in Israel in 1931 [21], with occasional outbreaks every few years [10]. Recently, five major outbreaks occurred in Israel, the first starting in 1999 among dairy cattle herds in the Jordan Valley, from which it then spread to the Mediterranean coastal plain [21]. A study conducted during the first outbreak evaluated the loss of an average of USD 112 cost per non-lactating cow and USD 280 per lactating cow mainly due to decreased milk production and increased abortion rate in infected herds [11]. The second outbreak started in 2004 and was much more widespread, covering most of Israel’s Mediterranean coastal plain; the third, in 2010, occurred in the interior plain. Studies conducted during and after this outbreak estimated an average net loss of 175.9 kg milk per cow affected by BEF and a total net loss from a single outbreak in a farm with an average of 250 lactating cows with approximately 40% morbidity up to USD 28,000 per herd [22,23]. The fourth outbreak occurred in 2014–15, and the disease was also recorded for the first time in the interior valley east of Haifa [2]. A study conducted during this outbreak estimated a 5 kg daily average reduction in milk production per cow that lasted 14 days. In dairy farms with a high percentage of infected cows (more than 20%), the economic loss was estimated at USD 5500. The fifth outbreak happened in the summer of 2017–8 and was recorded from the Jorden and Jezreel valleys, the Golan Heights, the interior valley east of Haifa, and the interior plain [20]. The sixth and most recent one occurred in 2021.

In Israel, farmers routinely report data on every BEF case to the Farm Herd Management Program (NOA), and they are registered to the Israeli herd book (Israel Cattle Breeders Association). In this study, we used the statistical capability of national data stored in the Israeli herd book to examine the effects of the 2021 outbreak on severely affected dairy farms nationwide. We tried to evaluate these economic losses in terms of the reduction in milk production, loss of valuable cows (culling rates), and abortion.

## 2. Materials and Methods

### 2.1. Infected Herds and Outbreak Distribution Countrywide

This study used retrospective data from the Israeli herd book (Israel Cattle Breeders Association). Farmers routinely report data on every BEF event to the Farm Herd Management Program (NOA). Retrospective data of reported BEF events were analyzed to present the accumulation and distribution of the number of new clinically diagnosed cases per day during the 2021 outbreak.

### 2.2. Economic Effect in Subsets of Highly Infected Dairy Farms

The data set examined in this section included thirty dairy farms that had a high proportion of infected animals. A dairy farm with a high proportion of infected animals was defined as a farm with at least fifty cows reported with clinical signs of BEF by the farm veterinarian in 2021, and the cows at the farm were confirmed positive for BEFV by qPCR [2,24]. A map of the farms included in this study is shown in Figure 1.

The parameters chosen to evaluate the economic effect were milk production loss, culling rates, and abortion rates.

i. Milk production loss was analyzed on cows with a daily milk yield from 30 days before BEF diagnosis until 30 days after diagnosis. Cows culled from the herd up to 30 days post BEF diagnosis was removed from the milk production analysis. Data were analyzed using SAS (version 9.2, SAS Institute, Cary, NC, USA). The GLM model was used to analyze milk production over time. The model includes the following effects: herd, the month of calving (January to December), the month of diagnosis (July to December), days from diagnosis (−30, 30), lactation number (1, 2, 3+), the interaction between days from diagnosis and lactation number and DIM, DIM^2^, and the square root (DIM) within lactation.

ii. To accurately evaluate culling rates due to BEF infection, only cows culled within ten days from their diagnosis date were included. The distribution of these days from the diagnosis date until culling was determined according to the farmers’ reports to the Farm Herd Management Program (NOA).

iii. Abortion rates were evaluated for cows that were pregnant during the BEF outbreak. They were analyzed using the Glimmix procedure in SAS. The model includes the following effects: herd, month/year (date) of insemination, lactation number (1, 2, 3+), and BEF diagnosed or non-diagnosed.

### 2.3. Economic Evaluations

According to the Israeli dairy board during the BEF outbreak (July–November 2021), the price of 1 kg of milk was USD 0.7, and the cost of an average cow was approximately USD 2300. Our economic evaluations were based accordingly.

## 3. Results

### 3.1. The 2021 BEF Outbreak

During this outbreak, severely infected herds were located in the following five geographic locations: the coastal plain (central and southern coastal strip), the Sharon plain (Northern coastal strip), the Negev desert, the upper Jordan valley (around the sea of Galilee) and lower Jordan valley (near the Dead Sea) (Figure 1).

The onset of the outbreak was at the end of July 2021. The number of cases remained relativity low until the beginning of September, when a rapid elevation was observed and reached a peak at the beginning of October with more than 300 infected cows per day nationwide. From October onward, the outbreak declined, reaching zero new cases at the end of December (Figure 2). According to the Israel Meteorological Service database, the temperatures along the Israeli coastal line between 27 July and 30 November 2021 were 29–24 °C, with 80–50% RH, and there were no rainfall events during that period (data retrieved from the Israel meteorological service database https://ims.gov.il/en/data_gov (accessed on 1 October 2023)).

### 3.2. Economic Losses in Affected Herds

The percentage of affected cows (morbidity) varied from 10% to 90.7%, with an average of 38.5% per herd, as shown in Table 1. Morbidity was extremely high in the Sharon plain (farms 25–27; 29 and 30 in Table 1). High morbidity was also documented in central Israel (farm 23), the Upper Jordan Valley (farm 24), and the Lower Jordan Valley (farm 28; Table 1).

#### 3.2.1. Milk Production

Figure 3A,B illustrates the reduction in milk production in BEF-infected cows. Cows exhibited a sharp reduction in milk production from 3 days before diagnosis until day 0 (day of diagnosis). In primiparous cows, milk yield was reduced by about 13.9 kg per cow per day (Figure 3A). The severe reduction in milk production lasted for approximately nine days (Figure 3A). In this period, the farm lost about USD 10 per cow per day (Table 2). This severe reduction in milk production lasted for approximately nine days, translating to a loss of USD 90 per infected cow during severe illness. In cows in their second lactation or higher, the reduction in milk production was even more significant, with a loss of about 15.9 kg in cows in their second lactation and 14.8 kg in cows in their third lactation or higher (Figure 3B). Again, the severe reduction in milk production lasted approximately nine days (Figure 3B). In this period, the farm lost about USD 100 per cow in their second lactation per day and about USD 93 per cow in their third lactation or higher per day (Table 2). All infected cows showed a slow improvement over time. It is important to note that even one month post-diagnosis, recovered cows did not return to the same level of production they had before their illness and produced one kg less per day compared to their production prior to diagnosis (Figure 3A,B). Thus, the losses in the month following infection were estimated to be an average of USD 315 per infected cow (Table 2).

#### 3.2.2. Culling Rates and Abortion Rates

The culling rates within ten days of BEF diagnosis ranged from 0 to 15.9%, with an average of 4.8% per herd (Table 1 and Figure 4). The average days from diagnosis with BEF to culling ranged between 0.7 and 7, with an average of 3.5 days (Table 1 and Figure 4). The highest mortality rates were documented in farms from the Negev desert (farm 1 in Table 1) and the coastal plain (farms 5 and 9; Table 1). The economic loss of one cow due to culling was, at the time of the outbreak, (according to the Israeli Dairy Board) USD 2300. Thus, in a small herd of 300 cows, for example, with an average culling rate, the economic losses were estimated to be USD 33,120, and in a herd of 1000 cows, the economic losses were estimated to be USD 110,400.

Abortion rates did not differ between cows in which BEF was not confirmed and cows in which BEF was confirmed (13.3% vs. 14.15%, respectively; *p*-value = 0.3744; Figure 5).

## 4. Discussion

This study evaluates the economic aspects of BEF outbreaks in dairy cattle herds due to reduced milk production, the loss of valuable cows (culling rates), and abortion. In order to conduct this analysis, we used the national data stored in the Israeli herd book to examine the effects of the 2021 outbreak on severely affected dairy farms countrywide. To our knowledge, this is the first extensive study on the impact of a BEF outbreak at a population level.

### 4.1. Characteristics of the 2021 BEF Outbreak

Figure 2 shows the time course of the outbreak, which lasted for more than four months, from the end of July until December. At the beginning of November 2021, more than 100 daily cases were still reported countrywide. It has been postulated that there is an association between the onset of heavy rainfall and the first clinical cases of BEF [9]. However, this was not the case in Israel, since the outbreak period between July and December 1st displayed high temperatures, high %HR, and no rainfall at all. In 2014, the BEF outbreak started in Israel during the summer, at the end of July (KVI annual reports). Our surveillance of sentinel, naïve dairy cattle showed that, from 2015, livestock in Israel were exposed to several arboviruses, including BEF, from early summer onward [20]. Therefore, it is evident that the last three BEF outbreaks in Israel (2014–2015; 2017–2018 and 2021) happened during the hot and dry summer season, and as global climate changes accelerate, we can expect more prolonged BEF outbreaks to occur in the Mediterranean basin.

Spatially, severe cases were distributed in five geographic locations as follows: the coastal plain (central and south of the coastal strip), the Sharon plain (Northern coastal strip), the Negev desert, the upper Jordan valley (around the sea of Gallie) and the lower Jordan valley (near the Dead Sea) (Figure 1). In former outbreaks in Israel, morbidity rates were estimated to be approximately 40%, with the highest morbidity and mortality rates documented from the Jordan Valley [10,22]. While this study exhibited similar results with an average of 38.5% morbidity rates nationwide, the geographical distribution differed as most herds with extremely high morbidity rates (more than 50%) were located in the Sharon plain. A spatial shift in culling rates was also observed during this outbreak, as discussed below (under Culling rates).

### 4.2. Reduction in Milk Production

It was shown that milk production usually drops by at least 50% in cows in which BEF has been diagnosed and that the highest-producing animals are generally the most severely affected. Milk yield should return to approximately 90% of previous levels after about three weeks, but cows affected late in lactation often do not return to production [9,10]. Our results show a reduction of more than 40% in diagnosed cows. Milk yield returned to approximately 90% of previous levels after more than 30 days. Israeli cows are the world’s highest-producing cows (https://my.icar.org/stats/list (accessed on 1 October 2023)). Although the percentage of the drop in milk production is lower than previous reports in high-producing cows, this reduction lasts longer, and the recovery of the cow is slower. As in previous reports, a sick cow does not truly recover from the infection as its milk production does not return to its original baseline, which not only causes immediate economic damage but also delayed damage, affecting the dairy farm income for months. Accordingly, during the first forty days from infection, the economic losses of the 2021 outbreak due to the reduction in milk production alone are estimated to be around USD 123,000 in a small herd of 300 cows and approximately USD 410,000 in a herd of 1000 cows.

### 4.3. Culling Rates

Deaths from ephemeral fever are uncommon and rarely involve more than 1–2% of the herd [7,9]. However, 5% to 10% losses have been recorded in severe and rare cases [8]. The 2021 outbreak was severe, with culling rates at an average of 4.8% per herd. Moreover, most cows were lost within four days of diagnosis. Former studies documented high mortality rates in BEF outbreaks in the Jorden Valley [10,22]. This outbreak was different. The highest culling rates (more than 10%) were recorded from three herds located in the Negev desert and the coastal plain. The herd with the highest culling rates of approximately 16% had almost 700 cows at the beginning of the outbreak (herd number 1 in Table 1). This herd suffered an estimated loss of USD 253,000 due to the loss of 110 cows. Its total loss from the outbreak (due to both morbidity and culling) was more than USD 275,000.

### 4.4. Abortion Rates

According to the literature, abortion occurs in approximately 5% of BEF-infected cows, especially those in the second trimester of pregnancy [25]. This study showed no significant difference between cows in which BEF was confirmed (14%) and cows in which BEF was not confirmed (13%). Nevertheless, our data show that the total abortion rates in all the herds tested were high. The high abortion rates in both groups could be attributed to prolonged and severe heat stress to which all cows in the Israeli herd, almost all of which are high-producing cows, are exposed for most of the year, especially during the hot, dry, long summer season in which the BEF outbreak took place [26]. In two farms with high morbidity rates (farms 27 and 29), blood-sucking (haematophagous) insects (mosquitoes and *Culicoides* biting midges) were collected for two consecutive nights during the outbreak (21 November). At the same time, whole blood was collected from six random cows on each farm. Insect pools and whole blood were tested for the presence of other arboviruses. BTV8 was detected by PCR [20] in both farms in insects and animals, and Akabane was detected via PCR [20] in insect pools. Although it is considered that a natural BEF virus infection can be prevented by an active infection of the bloodstream with another virus [25], our results suggest that the immune system of Israeli cows is under extreme stress from both acute climate conditions and the presence of multiple pathogens simultaneously, which may explain the high total abortion rates found in this study. In recent years, BEFV has been re-circulating extensively every three years and represents a constant threat to dairy and beef cattle in Israel [20]. Commercial attenuated and inactivated (killed) BEFV vaccines are available to Israeli farmers. Several studies have demonstrated that even though livestock vaccination against arboviral diseases can significantly reduce virus circulation [27,28], farmers’ willingness to use vaccines against viral diseases is relatively low worldwide [27,29]. On the one hand, this low level of willingness is surprising. Arboviral diseases such as BTV and SBV have been shown to harm the well-being and reproduction of farm animals severely and to cause significant financial damage to the dairy and meat industries over the past two decades [18,19]. Additionally, arboviral diseases such as bluetongue (BT) and epizootic hemorrhagic disease (EHD) that are listed as notifiable animal diseases by The World Organisation for Animal Health (WOAH) can cause substantial financial damage due to restrictions on animal trade and transportation like the current EHDV-8 outbreak in the Mediterranean basin affecting Italy, Spain and Portugal. On the other hand, high vaccine coverage against arboviruses is challenging to achieve due to the sporadic nature of their circulation [23,27]. Moreover, the costs of the time spent on gathering and restraining these animals and the actual costs of vaccines and the vaccination process are paid by the animal owners and significantly impact their willingness to vaccinate [27,29]. In the case of BEFV, the vaccines available to Israeli farmers also show low efficacy [23,28,30], which makes vaccination against this virus redundant for the time being. Consequently, in order to reduce arboviral diseases, including BEF, the Israeli Dairy Breeders’ Association and the Israeli Veterinary Services must combine efforts and “think outside the box” in the following two initiatives: first, consider changing their vaccination policy to support some of their costs and second, finding new ways to improve farm practices such as lowering animal populations in cow sheds, improving waste and water management to decrease the presence of *Culicoides* biting midges, mosquitos and other pests in the animal habitats and modify dairy farm infrastructures.

## 5. Conclusions

Arthropod-borne viruses (arboviruses) transmitted by blood-feeding insects such as mosquitoes and *Culicoides* biting midges are highly affected by climate changes. They are known to cause enormous economic damage as they affect herds’ morbidity, mortality, fertility, and abortion rates. In some cases, even the threat of infection can severely limit animal trade and transportation. This study demonstrates the substantial economic impact of BEF outbreaks in different parameters, such as a reduction in milk production and elevation in culling rates. Furthermore, this outbreak can have immediate and delayed effects, affecting the dairy farm’s income for months later. Globalization and climate change are increasing the risk of a BEF outbreak in Europe in the near future. Thus, the evaluations in this study enable the accurate risk assessment of BEFV emergence and emphasize the need for developing new vaccines and other effective strategies to fight arboviruses and their blood-sucking vectors.

## Figures and Tables

**Figure 1 vetsci-10-00645-f001:**
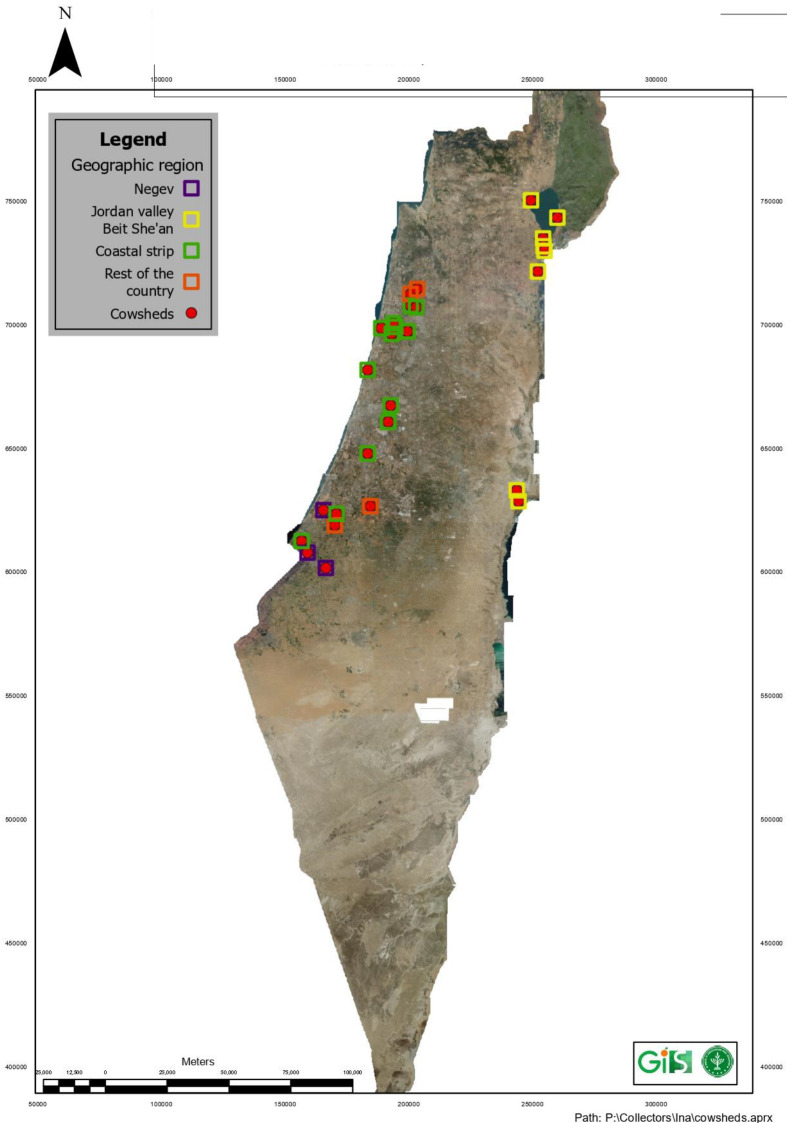
The geographic distribution of infected herds participating in the study throughout Israel.

**Figure 2 vetsci-10-00645-f002:**
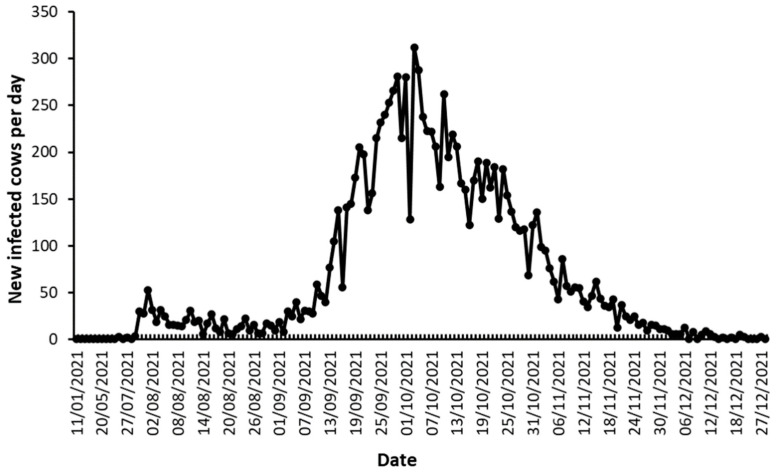
Outbreak duration and the daily number of newly BEF-clinical-diagnosed cows countrywide.

**Figure 3 vetsci-10-00645-f003:**
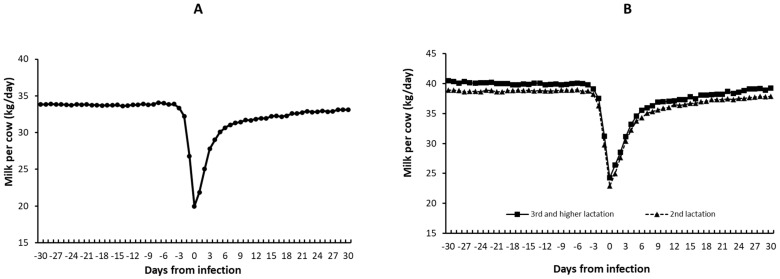
Changes in daily milk yield during the BEF outbreak. (**A**) Daily changes in primiparous cows. (**B**) Daily changes in multiparous cows.

**Figure 4 vetsci-10-00645-f004:**
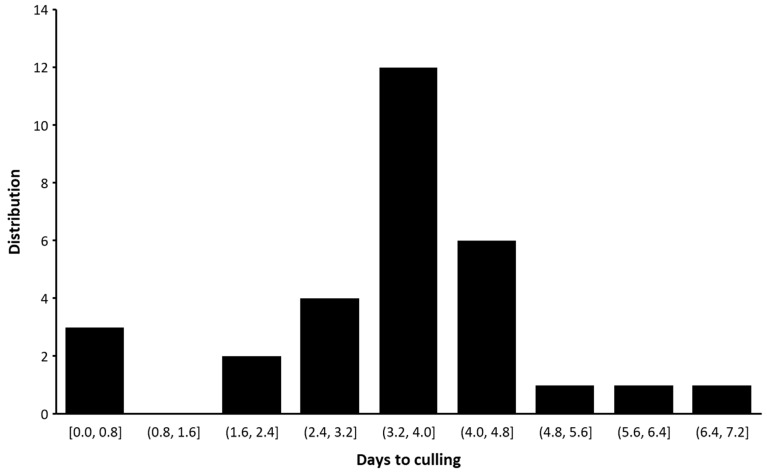
Distribution of days from the first clinical diagnosis of BEF until culling from the herd.

**Figure 5 vetsci-10-00645-f005:**
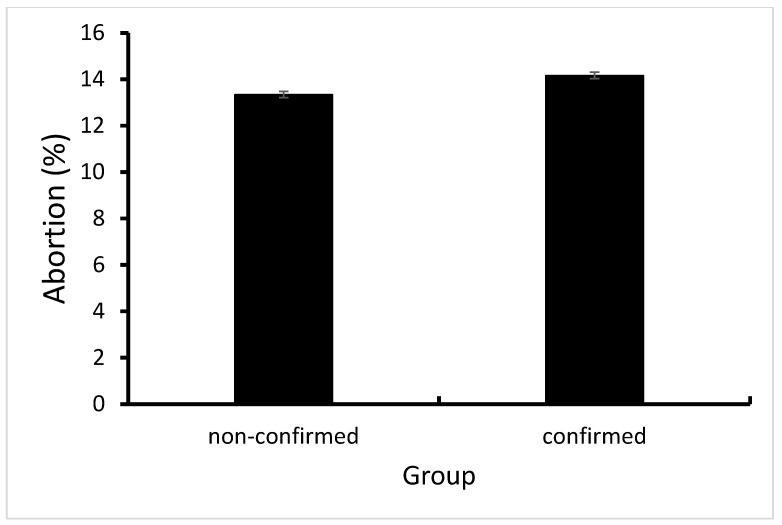
Abortion rate in cows in which BEF was not confirmed compared to cows in which BEF was confirmed.

**Table 1 vetsci-10-00645-t001:** Total number of cows, % morbidity and culling data of the farms participating in this study.

Farm	Average Number of Cows on the Farm during the BEF Outbreak	Number of Cows that Were Clinically Diagnosed with BEF during the Outbreak	(% Morbidity)	Number of Cows Culled within Ten Days of Diagnosis	Culling Rates (%) of BEF-Diagnosed Cows	Average Number of Days from Clinical Diagnosis to Culling
1	687	69	10.0%	11	15.9%	3.8
2	329	44	13.4%	2	4.5%	3.0
3	735	99	13.5%	3	3.0%	4.7
4	354	49	13.8%	0	0.0%	N/A
5	422	60	14.2%	8	13.3%	3.6
6	353	55	15.6%	3	5.5%	4.3
7	295	51	17.3%	0	0.0%	N/A
8	412	78	18.9%	6	7.7%	3.3
9	302	77	25.5%	8	10.4%	2.3
10	321	87	27.1%	4	4.6%	4.0
11	308	85	27.6%	2	2.4%	3.0
12	322	89	27.6%	8	9.0%	4.0
13	411	123	29.9%	7	5.7%	4.0
14	678	221	32.6%	13	5.9%	4.8
15	406	133	32.8%	5	3.8%	2.2
16	176	64	36.4%	1	1.6%	3.0
17	407	148	36.4%	3	2.0%	3.3
18	320	121	37.8%	3	2.5%	4.0
19	364	144	39.6%	8	5.6%	3.1
20	586	245	41.8%	11	4.5%	4.1
21	950	444	46.7%	22	5.0%	3.5
22	356	168	47.2%	2	1.2%	7.0
23	340	180	52.9%	3	1.7%	3.7
24	415	223	53.7%	3	1.3%	5.7
25	100	59	59.0%	3	5.1%	0.7
26	626	401	64.1%	20	5.0%	3.4
27	943	627	66.5%	52	8.3%	3.3
28	334	266	79.6%	11	4.1%	4.2
29	909	746	82.1%	25	3.4%	4.6
30	237	215	90.7%	4	1.9%	5.5

**Table 2 vetsci-10-00645-t002:** Economic loss from milk production in infected cows.

Lactation Number	Losses during Infection (9 Days) (USD per Cow)	Losses in the Month after Infection (USD per Cow)
1st	90	300
2nd	100	334
3rd and higher	93	311

## Data Availability

Essential data supporting the conclusions of this article are included in the main text.

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
