# Peer review of "Economic Aspects of Bovine Ephemeral Fever (BEF) Outbreaks in Dairy Cattle Herds"

_vetsci, 2023, doi:10.3390/vetsci10110645_

Round 1

Reviewer 1 Report

Comments and Suggestions for Authors

Overall, this manuscript is well written and the content should be published. While a number of authors have provided general overviews of the impact of BEFV infection on lactation in dairy cows, this is probably the first publication to extensively document the impact at a population level. There are a number of phrases that can be improved but these are largely editorial in nature. Once the authors have attended to these, the manuscript should be accepted for publication.

For ease of identification, the sections requiring attention have been marked on the original text extracted below and highlighted on the pdf file. Several of the figures need a more detailed caption, especially Fig 4.

Please  refer to both files attached to identify the sections that need to be edited

Comments on the Quality of English Language

Author Response

Reviewer 1:

Overall, this manuscript is well written and the content should be published. While a number of authors have provided general overviews of the impact of BEFV infection on lactation in dairy cows, this is probably the first publication to extensively document the impact at a population level. There are a number of phrases that can be improved but these are largely editorial in nature. Once the authors have attended to these, the manuscript should be accepted for publication.

For ease of identification, the sections requiring attention have been marked on the original text extracted below and highlighted on the pdf file. Several of the figures need a more detailed caption, especially Fig 4.

Please  refer to both files attached to identify the sections that need to be edited

A1: Check current taxonomy from ICTV

Answer: Done.  BEFV taxonomy was changed accordingly.

A2: Culicoides are not considered to play a role.

Answer: It is still debatable. A study from Pirbright published in 2020 claimed that BEFV multiplied only in Culicoides and not in mosquitos.  We published an article in 2023 that claim the both Culicoides and mosquitos can be potential vectors. Both references were added to the manuscript (references 4 and 5).

A3: Consider different expression - is the herd highly infected or does the herd have a high proportion of infected animals?

Agreed. We used “dairy farms which had a high proportion of infected animals”.

A4: GLM model used to analyze milk…- Something missing here?

Answer: Corrected to- “GLM model was used to analyze milk production over time.”

A5: “interaction between days from diagnostic and lactation number and DIM, DIM2, square root (DIM) within lactation” Change diagnostic to diagnosis.

Answer: Done.

A6: Define - I presume it is meant to be RH? (relative humidity)

Answer: True. The matter was corrected.

A7: Affected?

Agreed. Corrected.

A8: Please change - culling rate and mortality (deaths) are not the same

Answer: Agreed. Everything was changes to culling and culling rates.

A9: Please amend - perhaps cows in which BEF was confirmed or not?

Answer: Done.

A10: This text is an exact copy of the section in the abstract - please modify a little- probably edit the abstract.

Answer: Both the abstract and several paragraphs in the discussion were changed.

A11: Cows are not diagnosed - the disease is - pls amend "cows in which BEF has been diagnosed?

Answer: Agreed. It was corrected to: “in cows in which BEF has been diagnosed”

A12: Pls amend - if it was just PCR, then say so

Answer: Done. An appropriate reference was added.

A13: Results?.. Enable farmers… decision makers?

Answer: understood. Theis was deleted from the current version and the entire last paragraph of the discussion was changed.

Several of the figures need a more detailed caption, especially Fig 4.

Answer: All figure captions were amended.

Reviewer 2 Report

Comments and Suggestions for Authors

The work written by Lavon et al., and titled "Economic aspects of bovine ephemeral fever (BEF) outbreak on dairy cattle herds" describes the economic damage of BEV in Israel during a disease monitoring plan. The information described in this article is very useful and impactful (as well as worrying if this infection, as has already happened with other arboviruses, were to reach Europe). The manuscript is well written, although there are numerous formatting errors (line number is missing, references do not meet the Vet Sci criteria, ecc.) that prevent publication at this stage. Below are some of my comments aimed at improving, especially the introduction and discussion sections.

Abstract:

Authors wrote: “Bovine ephemeral fever (BEF) is a viral disease of dairy cattle which is suspected etc..”. Only dairy cattle? Only suspected?

What do the authors mean by “heavily conditioned animals”? Please change it.

The authors wrote: “Although BEF does not occur in the Americas and Europe (other than in the western regions of Turkey), the risk of BEFV emergence and spread in Europe is real. In the past two decades, arboviruses such as bluetongue virus (BTV) and Schmallenberg virus have emerged in Europe without warning through incursion routes that remain poorly defined. Since the European cattle population has never been exposed to BEFV, the economic losses to dairy and beef production in the continent may be considerable”. Too long for the abstract. This may be useful information for introducing arboviruses, but then the authors must write about their results, the importance of them, and the implications they have.

Change “power” to “potential” or something similar.

Introduction:

The introduction can be improved and more in-depth by talking about other arboviruses (such as SBV and BTV), which have moved from other countries to Europe with enormous damage. These viruses are still found today, having now become endemic (due to global warming). I report some bibliographical references that can help you explain this process and, in general, improve the references. doi: 10.1186/s13620-019-0147-3; https://doi.org/10.1186/s12917-023-03666-5; doi: 10.3168/jds.2023-23823; doi: 10.1111/1365-2664.13415; DOI: 10.3390/v11050412; DOI: 10.1016/j.prevetmed.2019.03.008; DOI: 10.1136/vr.104866; DOI: 10.1111/risa.14011; DOI: 10.1080/01652176.2020.1831708; DOI: 10.3389/fvets.2020.00065; DOI: 10.1093/jme/tjad098.

In general, the references in the text are incorrect. See Vet Sci guidelines.

Materials and Methods:

The authors wrote: “…new cases per day during the 2021 outbreak etc.” and “thirty highly infected herds”. Are the new cases and infected herds intended as clinical suspicions, or have they been tested via laboratory tests?

The authors should specify whether the confirmed cases confirmed a clinical suspicion or further testing. Also, which PCR test was performed (reference)? What sequence? What primers?

Results:

Please change the formatting in “3.2.1. Milk production”.

Please pull together the “Culling rates” and “Abortion rates” subsections.

Discussion:

Please delete the first part of the discussion (repetition), and I suggest removing the subsections for the discussion.

Please also improve the discussion with the most recent literature about the impact of arboviruses on the farm industry (where present).

Fig.5 5: The standard deviation is very high; please verify your data.

Comments on the Quality of English Language

Enghlish is fine and clear.

Author Response

Reviewer 2:

The work written by Lavon et al., and titled "Economic aspects of bovine ephemeral fever (BEF) outbreak on dairy cattle herds" describes the economic damage of BEV in Israel during a disease monitoring plan. The information described in this article is very useful and impactful (as well as worrying if this infection, as has already happened with other arboviruses, were to reach Europe). The manuscript is well written, although there are numerous formatting errors (line number is missing, references do not meet the Vet Sci criteria, ecc.) that prevent publication at this stage. Below are some of my comments aimed at improving, especially the introduction and discussion sections.

Abstract:

Authors wrote: “Bovine ephemeral fever (BEF) is a viral disease of dairy cattle which is suspected etc..”. Only dairy cattle? Only suspected?

Answer: Agreed. The sentence was changed to: ”Bovine ephemeral fever (BEF) is a viral disease of cattle which is transmitted by blood-feeding insects (mosquitoes and Culicoides biting midges).

What do the authors mean by “heavily conditioned animals”? Please change it.

Answer: Heavy cows (fat). Corrected accordingly.

The authors wrote: “Although BEF does not occur in the Americas and Europe (other than in the western regions of Turkey), the risk of BEFV emergence and spread in Europe is real. In the past two decades, arboviruses such as bluetongue virus (BTV) and Schmallenberg virus have emerged in Europe without warning through incursion routes that remain poorly defined. Since the European cattle population has never been exposed to BEFV, the economic losses to dairy and beef production in the continent may be considerable”. Too long for the abstract. This may be useful information for introducing arboviruses, but then the authors must write about their results, the importance of them, and the implications they have.

Answer: Agreed. The entire abstract was changed and an entire paragraph on the subject was added to the introduction.

Change “power” to “potential” or something similar.

Answer: Agreed . The sentence was changed to: ”we used the statistical capability of the Israeli national data stored in the Israeli herd book to examine…”

Introduction:

The introduction can be improved and more in-depth by talking about other arboviruses (such as SBV and BTV), which have moved from other countries to Europe with enormous damage. These viruses are still found today, having now become endemic (due to global warming). I report some bibliographical references that can help you explain this process and, in general, improve the references. doi: 10.1186/s13620-019-0147-3; https://doi.org/10.1186/s12917-023-03666-5; doi: 10.3168/jds.2023-23823; doi: 10.1111/1365-2664.13415; DOI: 10.3390/v11050412; DOI: 10.1016/j.prevetmed.2019.03.008; DOI: 10.1136/vr.104866; DOI: 10.1111/risa.14011; DOI: 10.1080/01652176.2020.1831708; DOI: 10.3389/fvets.2020.00065; DOI: 10.1093/jme/tjad098.

Answer: This is a very helpful comment and we would like to thank the reviewer for it. An entire paragraph on the subject was added to the introduction.

In general, the references in the text are incorrect. See Vet Sci guidelines.

Answer: Done.

Materials and Methods:

The authors wrote: “…new cases per day during the 2021 outbreak etc.” and “thirty highly infected herds”. Are the new cases and infected herds intended as clinical suspicions, or have they been tested via laboratory tests?

Answer: Clinically diagnosed. This information was added for clarity.

The authors should specify whether the confirmed cases confirmed a clinical suspicion or further testing. Also, which PCR test was performed (reference)? What sequence? What primers?

Answer: Agreed. Since we used previously published protocols, the proper references were added and the sentence was corrected from PCR to qPCR.

Results:

Please change the formatting in “3.2.1. Milk production”.

Answer: Done.

Please pull together the “Culling rates” and “Abortion rates” subsections.

Answer: Done.

Discussion:

Please delete the first part of the discussion (repetition), and I suggest removing the subsections for the discussion.

Answer: Agreed. The first and last paragraphs of the discussion were changed. We would prefer to keep the subsections in the discussion for clarity.

Please also improve the discussion with the most recent literature about the impact of arboviruses on the farm industry (where present).

Answer: Done.

Fig.5 5: The standard deviation is very high; please verify your data.

Answer: We would like to thank the reviewer for this comment. SD was checked and corrected.

Round 2

Reviewer 2 Report

Comments and Suggestions for Authors

Authors adressed my previous comments. 

Comments on the Quality of English Language

Minor editing of English language required